# Two forms of death in ageing *Caenorhabditis elegans*

Yuan Zhao[1,*], Ann F. Gilliat[1,*], Matthias Ziehm[1,2], Mark Turmaine[3], Hongyuan Wang[1], Marina Ezcurra[1], Chenhao Yang[1], George Phillips[1], David McBay[1], William B. Zhang[4], Linda Partridge[1,5], Zachary Pincus[4] & David Gems[1]

Ageing generates senescent pathologies, some of which cause death. Interventions that delay or prevent lethal pathologies will extend lifespan. Here we identify life-limiting pathologies in *Caenorhabditis elegans* with a necropsy analysis of worms that have died of old age. Our results imply the presence of multiple causes of death. Specifically, we identify two classes of corpse: early deaths with a swollen pharynx (which we call 'P deaths'), and later deaths with an atrophied pharynx (termed 'p deaths'). The effects of interventions on lifespan can be broken down into changes in the frequency and/or timing of either form of death. For example, *glp-1* mutation only delays p death, while *eat-2* mutation reduces P death. Combining pathology and mortality analysis allows mortality profiles to be deconvolved, providing biological meaning to complex survival and mortality profiles.

[1] Institute of Healthy Ageing, Department of Genetics, Evolution and Environment, University College London, London WC1E 6BT, UK. [2] European Molecular Biology Laboratory, European Bioinformatics Institute (EMBL-EBI), Wellcome Trust Genome Campus, Hinxton, Cambridge CB10 1SD, UK. [3] Department of Cell and Developmental Biology, University College London, London WC1E 6BT, UK. [4] Department of Genetics, Washington University in St Louis, St. Louis, Missouri 63110, USA. [5] Max Planck Institute for Biology of Ageing, Köln D-50931, Germany. * These authors contributed equally to this work. Correspondence and requests for materials should be addressed to D.G. (email: david.gems@ucl.ac.uk).

The nematode *Caenorhabditis elegans* is an excellent model organism for investigating the biology of ageing. Although much progress has been made in terms of identifying genes and pathways that affect lifespan[1,2], the underlying mechanisms of ageing remain poorly defined. One obstacle has been the difficulty of relating gene function to lifespan, given that the latter is a numeric, demographic parameter that contains little information about biological processes or structures to which gene function can readily be related. A complimentary approach is to study age-related pathologies and functional decline in relation to lifespan. As in humans, various senescent pathologies develop in ageing *C. elegans*, including deterioration of the pharynx, intestine, gonad and neurons[3–6]. While some pathologies do not appear to contribute to mortality[7], others may be life limiting. Identification of lethal senescent pathologies may provide us with the missing link between the biochemical function of gene products and their effects on lifespan.

In this study, we use necropsy analysis to investigate the causes of death in *C. elegans* and reveal two distinct modes of death, one that largely occurs earlier in life than the other. Thus interventions that alter lifespan in *C. elegans* reflect effects on timing and/or frequency of one or both types of death. We show how such differential effects can be resolved by mortality deconvolution, involving combined analysis of mortality and necropsy data.

## Results

**Necropsy analysis reveals two modes of death**. What do ageing *C. elegans* die of? To identify possible causes of death, we tracked pathologies in individual wild-type adult hermaphrodites as they aged (Supplementary Fig. 1; Supplementary Table 1) and tested for correlation between pathology severity and age at death. This revealed significant correlations between age at death and several pathologies, including pharyngeal deterioration (Fig. 1a; Supplementary Table 1). This, together with the previous observation that pharyngeal pumping span (that is, the length of time that the pharynx is active) correlates with lifespan[8], suggests that pharyngeal pathology could be life limiting.

Next, necropsy analysis was performed, for which corpses of nematodes that had expired from old age were collected daily and examined. This revealed two distinct types of corpse with respect to pharyngeal pathology (Supplementary Fig. 2). The first showed severe swelling of the posterior pharyngeal bulb, with a 20–120% increase in cross-sectional area (Fig. 1b). The second showed marked atrophy of the posterior bulb, with up to a 70% decrease in cross-sectional area (Fig. 1b). For convenience, we designated these corpse types 'P' ('big P') and 'p' ('small p'), respectively.

Notably, P deaths mainly occurred earlier than p deaths (Fig. 1c), with a median age of death (lifespan) of 12 and 22 days, respectively (Fig. 1d). The distinct timing contributes to the high variance in age at death seen in *C. elegans* populations despite their isogenicity[9,10], where >50% of the total variance can be explained by the existence of two types of death (Supplementary Table 2). In P deaths, pharyngeal swelling appeared only in the last few days prior to death (Fig. 1e). Swelling was preceded by a major reduction in pharyngeal pumping rate (Fig. 1f), likely contributing to the correlation between pharyngeal pumping span and age of death[8].

As in many animal species (and humans), *C. elegans* mortality rate increases with age. However, there is a hitherto unexplained deceleration of the age increase in mortality rate around day 10–12 (refs 11–13), postulated to reflect population heterogeneity in frailty[14]. The occurrence of this deceleration, which reflects a mid-life surge in death rate, was confirmed in the wild-type populations subjected to necropsy analysis in this study,

in which a slope change can be detected, with the most significant change on day 11 of adulthood (Fig. 1g; Supplementary Fig. 3a,b). The surge in mortality in mid-life was also seen in our archive mortality data collected at two locations (Supplementary Fig. 3c,d). In contrast, p mortality showed an exponential increase in mid-to-late life that, combined with the peak of P mortality in mid adulthood, leads to an apparent slowing of the mortality rate acceleration (Fig. 1h).

**Pharyngeal swelling is caused by bacterial infection**. Next, we explored the possible causes of P deaths, first asking: what is the immediate cause of pharyngeal swelling? The pharynx of immunocompromised *C. elegans* is susceptible to bacterial infection[15] and proliferation of the *Escherichia coli* food source limits worm lifespan[4,16]. Comparison of *E. coli* content in surgically excised pharynxes from live, aged worms found a 42-fold greater number of colony-forming units in swollen pharynxes compared to unswollen ones (Supplementary Fig. 4a), suggesting that the swelling is due to increased bacterial content. To visualize localization of bacteria within pharyngeal tissue, we fed worms with *E. coli* expressing red fluorescent protein (RFP). Red fluorescence was seen throughout the pharyngeal tissue in worms that undergo P death (Fig. 2a), whereas p corpses typically contained no fluorescence or only small fluorescent inclusions in the posterior bulb, perhaps reflecting contained invasions (Fig. 2b; Supplementary Fig. 4b). Live worms in the early stages of bacterial invasion revealed RFP co-localized with green fluorescent protein (GFP) markers of several different pharyngeal cell types but most often with pharyngeal muscle near the grinder (Fig. 2d; Supplementary Fig. 4c,d). Moreover, examination of swollen pharynxes using transmission electron microscopy (TEM) showed them to be filled with bacteria (Fig. 2c), while unswollen pharynxes contained either no invading *E. coli* or small, membrane-bound bacterial inclusions usually near the grinder cuticle (Supplementary Fig. 5). These results suggest a route for initial bacterial invasion through the pharyngeal cuticle in the grinder region.

To test whether pharyngeal swelling is caused by bacterial proliferation, *E. coli* were treated with an antibiotic (carbenicillin) or ultraviolet irradiation. In each case, P death was eliminated (Fig. 2e), and lifespan extended (Supplementary Fig. 6a,b; Supplementary Table 3) as observed previously[4,16]. As expected, blocking bacterial proliferation also removed the mortality rate deceleration seen around day 11 (Fig. 2f); this is also consistent with the absence of mid-life mortality deceleration in populations maintained in the recently described automated systems, including worm corrals[17] and lifespan machines[18], as P death is either eliminated or significantly reduced in both cases (Supplementary Fig. 7). However, elimination of P death did not completely abrogate the correlation between pharyngeal pumping span and lifespan (Supplementary Fig. 6c,d), which could imply that pharyngeal pathology contributes to mortality in additional ways.

**High pharyngeal pumping rate promotes P death**. Thus P death is associated with pharyngeal invasion and proliferation of bacteria, but why do these deaths occur relatively early, with the majority occurring before day 15? Shifting worms raised on non-proliferating bacteria to proliferating bacteria at time points after day 4 progressively reduced the frequency of P death and reduced early mortality (Fig. 3a,b; Supplementary Fig. 8a–d; Supplementary Table 4), suggesting that a narrow time window exists in early adulthood where worms are susceptible to pharyngeal infection. This, together with initial occurrence of invasion near the grinder, could imply that the high rate of pumping in

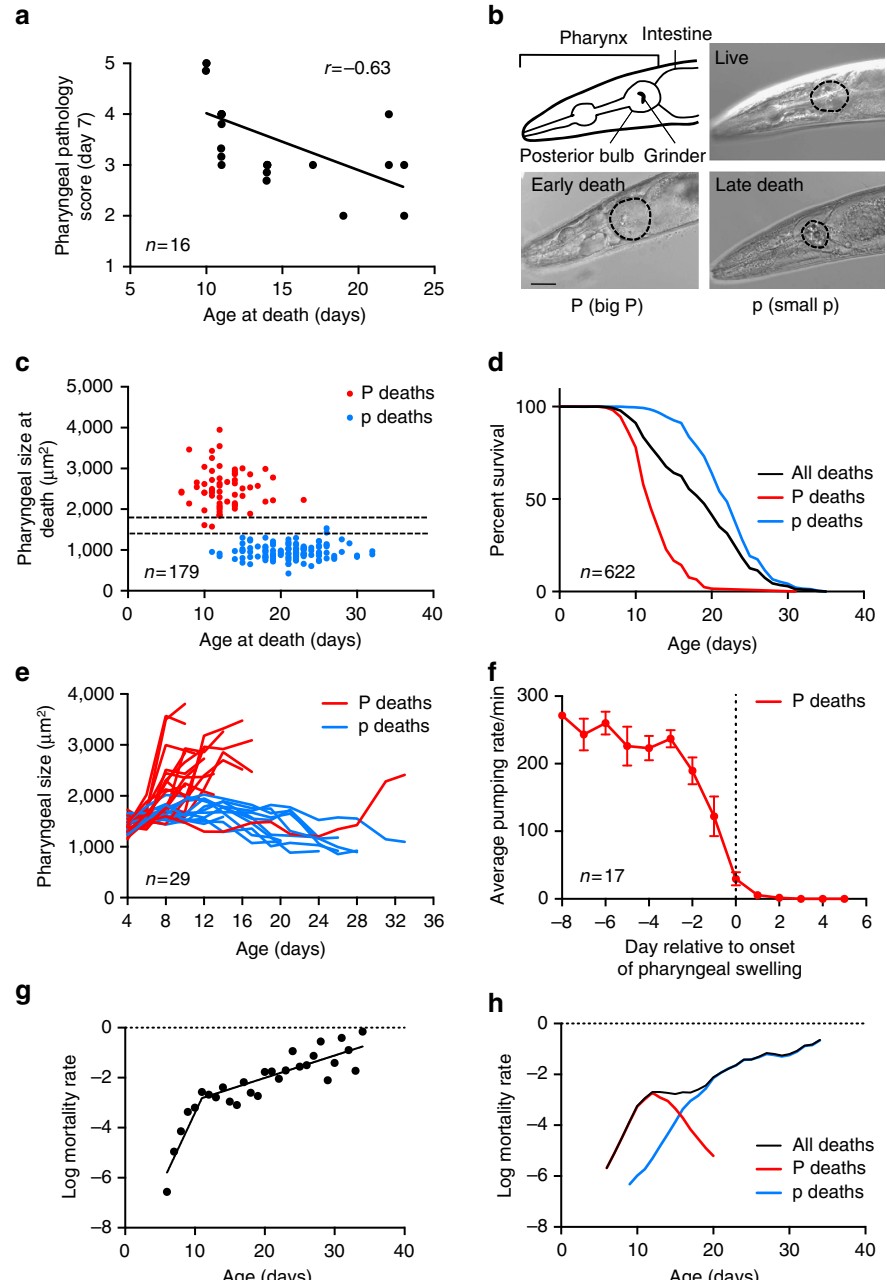

**Figure 1 | Two types of corpse in ageing *C. elegans* populations.** (**a**) Positive correlation between pharyngeal pathology on day 7 of adulthood and age at death. ($n = 16$). (**b**) Corpses with an enlarged or atrophied pharynx. For comparison, the pharynx of a healthy 10-day-old adult worm is shown. The posterior bulb is outlined. Scale bar, 40 μm. (**c**) Age distribution of P and p deaths. y Axis shows cross-sectional area of pharynxes on the day of death ($n = 179$). Dotted lines represent the size of a healthy pharynx on day 10 (mean ± s.d.). (**d**) Survival curves of all worms (black) or with P and p deaths resolved (red and blue, respectively). Data compiled from 11 independent trials ($n = 622$). An additional single large trial ($n = 587$) was also performed (Supplementary Fig. 3e). (**e**) Age changes in pharyngeal size in individual worms. Early deaths (mostly pre-day 15) are preceded by a dramatic increase in pharyngeal size. The pharynxes of the remaining worms gradually decrease in size until death ($n = 29$). (**f**) Mean pumping rate of worms that die with P. Day 0, first day of swelling. Worms show a major reduction in pumping rate immediately before pharyngeal swelling. Data are mean ± s.e.m. (Trials: 1, $n = 17$). (**g**) Segmented linear regression analysis on log mortality rate of wild-type survival under standard conditions (data as in **d**, $X_0 = 11$). The slopes before and after day 11 are significantly different (slope test $P = 2.81 \times 10^{-9}$). (**h**) Mortality deconvolution of wild-type survival (data as in **d**). Age-specific mortality rates of the whole population (black) or with P and p deaths resolved (red and blue, respectively).

young adults[8] injures the pharyngeal cuticle, perhaps due to mechanical stress, causing cuticular perforations and vulnerability to invasion. If this is correct, then reducing pharyngeal pumping rate should suppress P deaths. We therefore examined a range of pumping defective mutants, including *eat-2(ad1116)* mutants, which have a reduced pumping rate[19]. In most cases, P deaths were reduced or eliminated (Fig. 3c). This suggests that a wild-

type, high rate of pharyngeal pumping leads to bacterial invasion of pharyngeal tissue.

The longevity of *eat-2* mutants has previously been attributed to dietary restriction[20]. However, results of mortality and necropsy analysis indicated that the *eat-2* lifespan extension observed was largely attributable to reduction in the frequency of P death, since the lifespans of P and p subpopulations did not

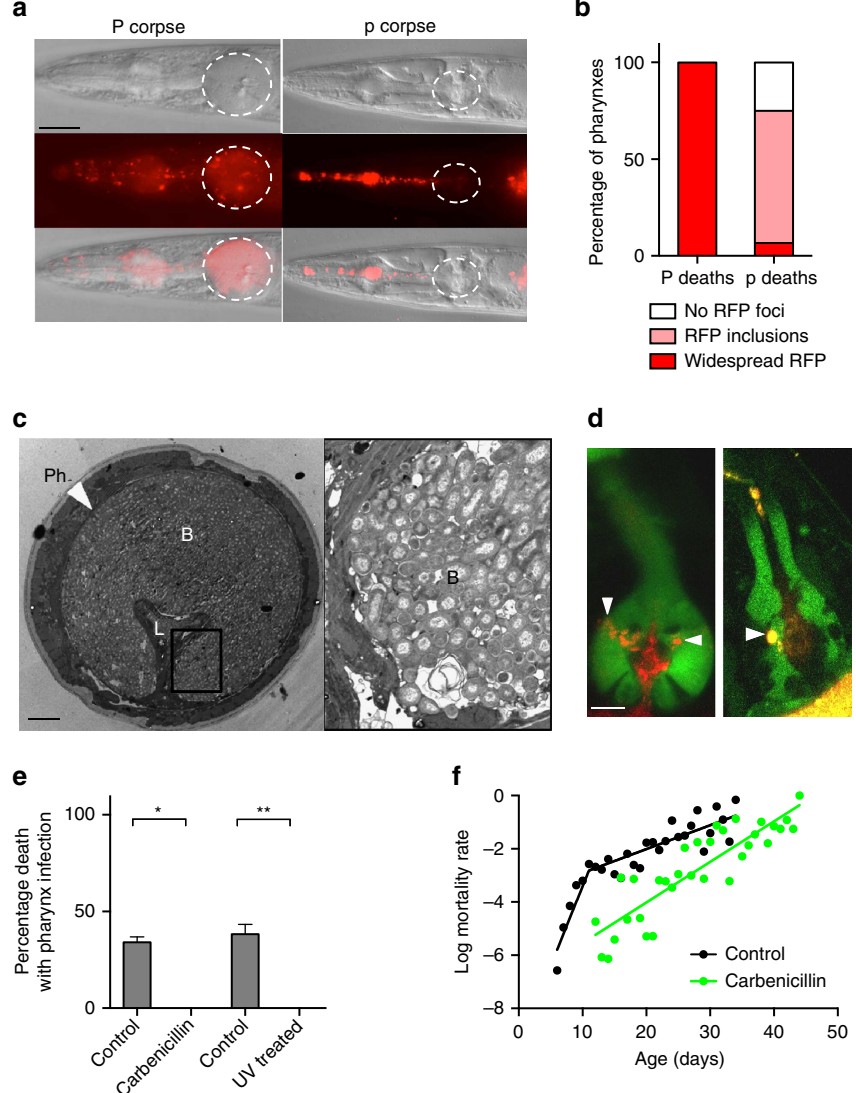

**Figure 2 | Swelling in P deaths results from *E. coli* infection.** (**a**) Comparison of P and p corpses of worms fed with *E. coli* expressing RFP. Scale bar, 40 μm. (**b**) Proportion of P ($n = 70$) and p ($n = 75$) corpses with widespread RFP, small intra-pharyngeal RFP inclusions or no detectable RFP foci. (Trials: 2). (**c**) Representative TEM of a swollen pharynx from a live, 8-day-old adult. L, lumen; Ph, pharynx; B, bacteria. The elongated appearance of some bacilli suggests that proliferation occurs within pharyngeal tissue. For TEMs of pharynxes at different stages of infection, see Supplementary Fig. 5. Scale bar, 5 μm. (**d**) Early stages of *E. coli*-RFP invasion of pharynx in transgenic *C. elegans* expressing GFP in selected cell types within the pharynx. Representative images of the muscle (left) and marginal cell marker (right) are shown with nascent intra-pharyngeal infection near the grinder (arrowheads). *E. coli* were noted previously in senescent marginal cells[31]. For gland cell marker image, see Supplementary Fig. 4c. Scale bar, 10 μm. (**e**) Blocking *E. coli* proliferation prevents P death. Proportion of P corpses when worms are grown with or without proliferating *E. coli*. Data are mean ± s.e.m. (Carbenicillin trials: 4; ultraviolet trials: 2, for sample size, see Supplementary Table 3). (**f**) Carbenicillin-treated bacteria remove mortality rate deceleration in mid life. Control wild-type mortality rate (black) as in Fig. 1g. For carbenicillin-treated log mortality rate (green), the slopes of regression are not significantly different when segmented at any point.

significantly differ between N2 and *eat-2* (Fig. 3d; Supplementary Table 5). This suggests, against expectation, that the longevity of *eat-2* mutants is attributable, at least in part, to suppression of P death rather than to dietary restriction.

**Wound healing may protect against bacterial infection.** Why do only a proportion of worms undergo P death? One possibility is that some worms have a higher pharyngeal pumping rate during early adulthood, resulting in sufficient damage to the cuticle to allow bacterial invasion to occur. However, in early adulthood there was no difference in pumping rate between worms that subsequently underwent P versus p death (Supplementary Fig. 8e). This argues against predisposition due

to intrinsic differences in pumping rate. Another possibility suggested by the early adult time window for predisposition to P death is that early mechanical injury to the pharyngeal cuticle subsequently heals, preventing further infection in the majority of worms; the *C. elegans* external cuticle has an efficient wound healing capacity[21]. TEM of the grinder region in older worms revealed major cuticular scars (Supplementary Fig. 9), similar in appearance to cuticular scars described previously[21], consistent with injury and subsequent healing. Together these results suggest that rapid pumping in early adulthood leads to mechanical damage to the pharyngeal cuticle, allowing initial invasion of *E. coli*. In some worms, subsequent proliferation of invading *E. coli* leads to pharyngeal infection and P death, while in others

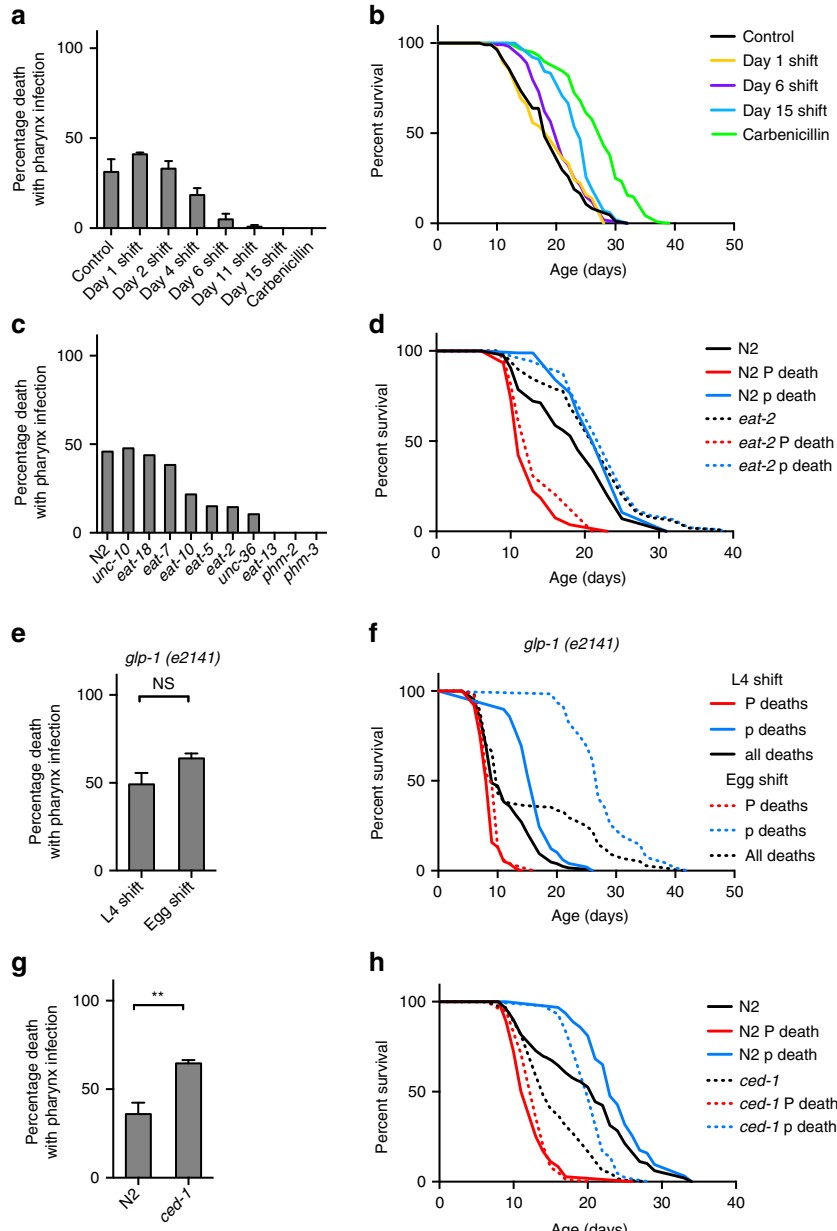

**Figure 3 | High pharyngeal pumping rate promotes P death.** (**a**) Effect of maintenance on non-proliferating *E. coli* (carbenicillin) during early life on frequency of P deaths. Exposure to proliferating bacteria in early-mid adulthood is required for P death to occur. (Trials: 2, for sample size, see Supplementary Table 4). (**b**) Maintenance on non-proliferating *E. coli* (carbenicillin) during early life reduces early mortality (data as in **a**). (**c**) Effect of pumping-defective mutants on the frequency of P. (Trials: 1–2). (**d**) Effect of *eat-2(ad1116)* on lifespan. Whole population (black) or P and p deaths resolved (red and blue, respectively). Log-rank test between N2 and *eat-2* whole populations $P = 0.0007$, P death subpopulations $P = 0.1996$, p death subpopulations $P = 0.0815$. (Trials: 2, for sample sizes, see Supplementary Table 5). (**e**) Germline defective mutant *glp-1(e2141ts)* increases lifespan without affecting P death frequency when shifted to 25 °C as eggs. (Trials: 2, for sample sizes see Supplementary Table 6). (**f**) Effect of *glp-1* on lifespan, when shifted to 25 °C at either egg (dotted) or L4 larval stage (solid). Median lifespan: L4 shift P deaths 9 days, p deaths 17 days; egg shift P deaths 9 days, p deaths 27 days (data as in **e**). (**g**) *ced-1(e1735)* mutant increases the frequency of P deaths. (Trials: 2, for sample sizes, see Supplementary Table 7). (**h**) Effect of *ced-1* (dotted) on lifespan compared to wild-type (solid). Median lifespan: *ced-1* P deaths 13 days, p deaths 20 days; wild-type P deaths 13 days, p deaths 23 days (data as in **g**).

invasion remains contained after cuticular healing (Supplementary Fig. 10). The reason behind such heterogeneity remains to be discovered but could reflect differences in burden of initial invasion or in the ability to heal or to fight invasion.

**Differential effects on P and p in long-lived mutants.** In this study, we have shown how combined pathology and mortality analysis allows deconvolution of *C. elegans* mortality profiles,

providing a means to understand how interventions alter lifespan in terms of effects on pathology. Examples modelling the effects on mortality and survival due to theoretical alterations in frequency or timing of P and/or p deaths are shown in Fig. 4, many of which resemble real lifespan data observed here and in previous studies (Supplementary Fig. 11). For example, lack of a germline in *glp-1(e2141)* delays p deaths by >10 days but has little effect on either the timing or frequency of P death (Fig. 3e,f;

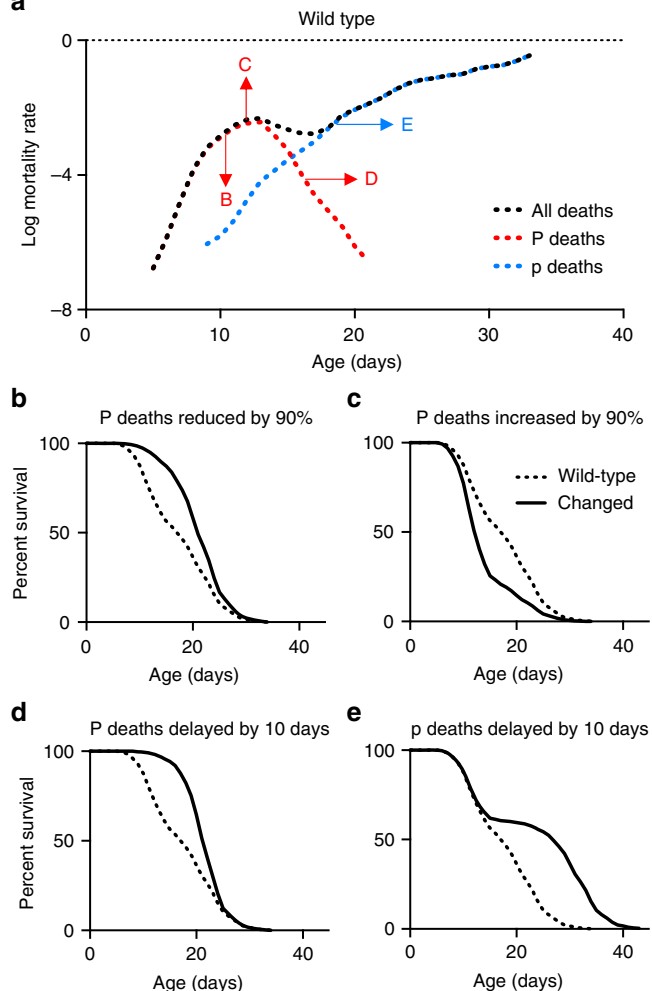

**Figure 4 | How mortality deconvolution can decipher survival curves.**
(**a**) Log mortality rate of an idealized wild-type population. P and p survival
were adapted from the P and p death survival calculations in Fig. 1g, and the
survival proportions of the whole population were calculated, assuming that
P death is 40% of the population and p death is 60%. The resulting survival
data were used to calculate the log mortality rate. Arrows show the
direction of mortality rate shift by transformations of survival data resulting
in lifespan changes shown in (**b**–**e**). Dotted lines, wild-type data; solid lines,
data after specified transformation. (**b**) 90% decrease in P deaths. (**c**) 90%
increase in P deaths. (**d**) 10-day delay in P deaths. (**e**) No change in P
deaths, with 10-day delay in p deaths. For mortality rate plots for (**b**–**e**), see
Supplementary Fig. 11.

Supplementary Table 6), which explains the atypical neck-and-
shoulder shape of its survival curve[22]. Another example is the
short-lived *ced-1(e1735)* mutant, which is hyper-susceptible to
pharyngeal invasion by pathogenic bacteria[15]. Its short lifespan
on *E. coli* is due to increased frequency (without changes in
timing) of P death, as well as faster die-off of p worms (Fig. 3g,h;
Supplementary Table 7). In contrast, *eat-2* provides an example of
how increased lifespan can result from reduced P death frequency
(Fig. 3d). Thus different lifespan-altering interventions can act by
differentially affecting different pathologies.

## Discussion

As a model organism, *C. elegans* has generated numerous insights
and discoveries in the ageing field that have proven to be
applicable in higher organisms. However, a long-standing
obstacle is that it is difficult to follow the causal chain connecting
genes and proteins to demographic parameters, such as lifespan
and mortality rate. We argue that to better understand ageing in
*C. elegans* requires knowledge not just of genes that influence
lifespan but also of the pathologies of ageing and how these cause
death. This study describes a novel approach to understanding
demographic data in *C. elegans* based on the combined analysis of
pathology and mortality (and conceptually similar approaches
have been used in human clinical studies, c.f. the competing risks
model[23]). This provides a route to discover the biological
mechanisms of ageing underlying survival curves and mortality
profiles in this powerful model organism.

But is P death attributable to ageing, or is it merely the result of
bacterial infection? In order to maximize the contribution of
intrinsic determinants of senescence, would it not be better
simply to exclude P death, for example, by using antibiotics?
Certainly, excluding P death can facilitate analysis of intrinsic
determinants of senescence. However, our results suggest that the
P subpopulation is itself a useful model for studying the biology
of ageing. For example, they suggest that damage incurred early
(mechanical damage to the pharyngeal cuticle and contained
infections within pharyngeal tissue) lie latent during early–mid
adulthood but then recrudesce in later life, potentially due to
action of intrinsic determinants of senescence (Supplementary
Fig. 8e). Thus the P subpopulation may be used to investigate
how early damage can interact with later intrinsic causes of
ageing to determine senescent pathology and mortality.

## Methods

**Culture methods and strains.** Standard *C. elegans* maintenance was performed
using standard protocol[24]. Strains were grown at 20 °C on nematode growth media
(NGM) plates seeded with *E. coli* OP50 to provide a food source. *C. elegans* strains:
An N2 hermaphrodite stock recently obtained from the Caenorhabditis Genetics
Center was used as wild type (N2 CGCH)[25]. For necropsy analysis: CB3203 *ced-
1(e1735) I*, CB4037 *glp-1(e2141) III*, DA597 *phm-2(ad597) I*, DA1116 *eat-2(ad1116) II*,
DA493 *phm-3(ad493) III*, DA522 *eat-13(ad522) X*, DA698 *unc-36(ad698) III*,
DA521 *egl-4(ad450) IV*, DA606 *eat-10(ad606) IV*, DA464 *eat-5(ad464) I*, DA591
*unc-10(ad591) X*, DA1110 *eat-18(ad1110) I*. For confocal microscopy: BC12677
*dpy-5(e907) I*; *sIs11111 [rCesC32F10.8::GFP + pCeh361]* (GFP in muscle cells),
BC12754 *dpy-5(e907) I*; *sIs12567[rCesC07H6.3::GFP + pCeh361]* (GFP in marginal
cells), BC16329 *dpy-5(e907) I*; *IsEx16329[rCesF20B10.1::GFP + pCeh361]* (GFP in
g1 pharyngeal gland cells).

*E. coli* OP50 expressing red fluorescent protein (OP50-RFP) was generated by
transforming OP50 with plasmid pRZT3 (kindly provided by J.F. Rawls, Duke
University, originally from W. Bitter, Vrije University)[26]. pRZT3 contains genes for
tetracycline resistance and RFP (DsRed), under the control of a constitutive *lac*
promoter. OP50-RFP was grown on lysogeny broth (LB) plates and in LB broth in
the presence of 25 µg ml$^{-1}$ tetracycline. However, before NGM plates were seeded,
OP50-RFP was resuspended in LB broth without tetracycline. This was to avoid
effects of tetracycline on bacterial pathogenicity that might occur even in the
presence of the tetracycline-resistance plasmid. Worms were then raised from
hatching on OP50-RFP.

To examine the role of proliferating bacteria on pharyngeal swelling, bacterial
growth was prevented in two ways. Carbenicillin was added topically onto a 2-day-
old bacterial lawn to a final concentration of 4 mM. Unless otherwise specified,
worms were transferred from untreated OP50 at L4 stage. For ultraviolet killing,
80 µl of OP50 was added to each NGM plate and left overnight at 20 °C. Plates
were then exposed to ultraviolet light for 30 min. Worms were raised on ultraviolet-
killed bacteria from aseptic eggs.

**Lifespan and Ziehm tables.** Lifespan measurements were performed as follows.
Trials were conducted at 20 °C unless otherwise specified and without 5-fluoro-2′-
deoxyuridine (FUDR). Worms were transferred daily during the reproductive
period. Mortality was scored daily to ensure that corpses were fresh and not
decomposed, to aid necropsy analysis. Worms that were scored as dead were
collected for microscopy. Lifespan measurements were repeated at least twice
unless otherwise stated, and survival plots were generated using the combined
lifespan data.

In addition to standard presentations of survival data in the form of survival
curves and tables of analysed statistics (for example, mean lifespan, comparison
statistics), we have also included tables of raw mortality data. The purpose of the
latter is to enable future investigators to reanalyse the data, for example, in data
mining approaches or for the purpose of resolving discrepancies between different
published reports. To this end, we employ a mortality data reporting format
developed by M. Ziehm, which applies minimal reporting standards to ensure

inclusion of all essential information about experimental conditions. Data include specification of causes of data censoring, as follows: bag = bag of worms (death from internal hatching of eggs); cont = contaminated (bacterial or fungal infection); dis = disappeared (investigator failed to locate worm); kil = killed (worm accidentally killed during handling); cen = censored (censored without a recorded reason); otw = on the wall (worms climbed wall of Petri dish and died from desiccation); and rup = rupture (internal organs extensively extruded through vulva). We include, for other investigators who choose to take this approach, an unpopulated Ziehm table (Supplementary Data 1).

**Survival and mortality analysis.** For survival analysis of P or p deaths only (Fig. 1d), all observations from worms with pharynx pathology of the other type were excluded. For mortality analysis (Fig. 1h), log mortality rate of the P and p cohorts were calculated with the other type censored, which allows presentation of the risks in relation to the whole population. Survival and mortality plots were generated using GraphPad Prism with no smoothing.

For the archive data collected at University of Missouri-Columbia (1994–1996) and UCL (1998–2000), mortality plots were generated using SurvCurv[27] using a sliding window smoothing of ± 2. Analysis of mortality data revealed that for unknown reasons N2 worms lived 3 days longer at UCL than at UMC. Therefore, we adjusted for this 3-day difference when combining and plotting the mortality data.

**Microscopy.** Live worms or corpses were mounted onto 2% agar pads. Live worms were anaesthetized by placing them in a drop of 0.2% levamisole. Nomarski images of worms were collected using a Zeiss Axioskop 2 plus microscope with a Hamamatsu ORCA-ER digital camera C4742-95 and Volocity 6.3 software (Macintosh version) for image acquisition. The presence of *E. coli* OP50-RFP in the pharynx was assayed using a rhodamine filter cube (excitation: 540–552 nm; emission: > 590 nm).

Before confocal or electron microscopy, aged worms were separated into three groups based on the pattern of red fluorescence from *E. coli* OP50-RFP in their pharynxes: a pharynx full of RFP was taken to be fully infected; a pharynx with medium-sized RFP inclusions was taken to be at an early stage of infection; and a pharynx with no RFP was taken to have no infection. Confocal images were acquired using Zen2009 software driving a LSM-710 confocal station with an inverted Zeiss Axio Observer microscope.

For electron microscopy, worms were prepared using a standard protocol (protocol 8 (ref. 28)). In brief, animals were washed in M9 and then cut in half using 30 G needles. Heads were collected in fixing solution (2.5% glutaraldehyde, 1% paraformaldehyde in 0.1 M sucrose, 0.05 M cacodylate) on ice, rinsed 3 times in 0.2 M cacodylate, fixed in 0.5% $OsO_4$ and 0.5% $KFe(CN)_6$ in 0.1 M cacodylate on ice and sequentially washed in 0.1 M cacodylate and 0.1 M NaAcetate. Worms were then stained in 1% UAc in 0.1 M NaAcetate (pH 5.2) for 60 min, rinsed 3 times in 0.1 M NaAcetate and rinsed overnight in distilled water. Samples were then embedded in 3% seaplaque agarose, dehydrated and infiltrated through ethanol and propylene-resin series and then cured in 60 °C oven for 3 days. Serial 1 µm sections were taken for light microscopy and at the region of interest ultra-thin sections were cut at 70–80 nm using a diamond knife on a Reichert ultramicrotome. Sections were collected on 300 mesh copper grids and stained with lead citrate before being viewed in a Joel 1010 transition electron microscope. Images were recorded using a Gatan Orius camera. Images were obtained using the Gatan imaging software and then exported into TIFF format.

**Measuring *E. coli* content of the pharynx.** Worms on day 10 or 11 of adulthood (20 °C) were divided into two groups, those with and without pharyngeal swelling. The posterior bulb of the pharynx was then dissected out using a 31 G needle. These were placed in 120 µl M9 buffer and macerated using the needle. The resulting lysates were vortexed for 30 s and then centrifuged for 1 min at 13,000 r.p.m. Bacterial concentration in supernatants was determined after serial dilution in M9 solution by plating onto LB plates, followed by counts of colony-forming units.

**Necropsy analysis.** A series of microscope images at × 400 or × 630 magnification were collected along the length of each corpse using Nomarski optics and examined for the presence of any unusual pathologies beyond those typically seen in elderly hermaphrodites[4,5,29]. The pharynx in particular was examined closely to determine whether bacteria was visible within the tissue, and then the area of the posterior bulb was measured using either the Image J or the Volocity 6.3 software.

**Measuring pharyngeal pumping rate.** Worms were examined *in situ* on NGM agar plates using a Nikon SMZ645 microscope for 15 s, and the number of pharynx pumps was scored manually using a clicker counter. This was repeated three times for each worm, from which mean pumps per minute was calculated.

**Longitudinal pathology analysis on individual worms.** Worms were cultured individually at 20 °C. On days 4,7,11 and 14 of adulthood, each worm was imaged individually. For imaging, microscope slides were prepared by taping two cover-slips on the slide, at each edge, leaving an empty space in the middle for the agarose

pad. The worm was then placed on a 2% agarose pad on the slide. Another coverslip was then placed on top but resting on the two side coverslips to reduce the pressure exerted on the worm. The slide was then placed on a PE120 Peltier cooling stage (Linkam Scientific) set to 4 °C. Within a couple of minutes of cooling, the nematodes ceased to move, and images were taken at × 630 magnification using a Zeiss Axioskop. After imaging, each worm was carefully recovered by pipetting 20 µl of M9 buffer between the top coverslip and the agar pad. The coverslip was lifted gently to minimize stress to the worm. The worm was then returned to an NGM plate. Images of pharynxes, distal gonads and uterine tumours were scored on an ordinal scale with scores 1–5 (refs 4,7). Score 1 represents a healthy tissue; score 2 indicates some minor tissue disorder; scores 3/4 are where minor/major pathology has developed and score 5 is where pathology development has reached its maximum. Yolk pools and intestinal atrophy were analysed using semi-quantitative approaches, measuring the area of the yolk pools and intestinal width relative to body area or width, respectively. Lifespans were recorded every day.

**Statistics.** For lifespan assays, survival and mortality graphs were generated using the Graphpad Prism software, and statistical tests for significant difference between survival curves were performed using the log-rank test. Log-rank tests of both the P cohort versus the p cohort and the P cohort versus all deaths showed highly significant differences ($P < 10^{-15}$ and $P < 10^{-6}$, respectively), and the p cohort is also significantly different ($P = 10^{-3}$) from the cohort of all deaths. For analysis of mortality deceleration, the significance of slope change was tested using GraphPad Prism comparing slopes of linear regression lines[30]. Correlations from single worm, longitudinal pathology analysis were analysed using the Spearman Rank test. The correlation between pumping span and age at death was calculated using a linear regression model, and the comparison of these slopes between worms either on proliferative or non-proliferated bacteria were calculated using a linear model in JMP 11.

**Data availability.** The data that support the findings of this study are available from the corresponding author upon reasonable request. For the survival assays we have shown, raw data are available in the form of Ziehm tables provided in Supplementary Data 1 of this publication.

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

## Acknowledgements

We thank N. Alic, A. Benedetto, F. Cabreiro, N. Pujol (Marseille University) and members of the Gems lab for useful discussion and/or comments on the manuscript; J.F. Rawls (Duke University) and N. Stroustrup (Harvard University) for reagents; and C. Au for technical assistance. We also thank D. Hall and L. Herndon (Albert Einstein College of Medicine) for advice and access to TEM images on WormImage.org, which is supported by NIH (OD 010943). Some strains were provided by the Caenorhabditis Genetics Center, which is funded by NIH Office of Research Infrastructure Programs (P40 OD010440). This work was supported by a Wellcome Trust Strategic Award (098565/Z/12/Z) and an EU grant (FP6-518230, IDEAL).

## Author contributions

A.F.G., D.G. and Y.Z. conceived the project. A.F.G., M.E., D.M., L.P., Z.P., H.W., C.Y., W.B.Z. and Y.Z. performed experiments and/or analysed and/or interpreted data. M.T. prepared worms for electron microscopy. G.P. and M.Z. analysed archive lifespan assay data and mortality data. A.F.G., D.G. and Y.Z. wrote the paper.

## Additional information

**Competing interests:** The authors declare no competing financial interests.

