## [Peer Review File · Nature Communications]

Reviewers' comments:

Reviewer #1 (Remarks to the Author):

The manuscript by Zhao et al aims at stratifying *C. elegans* mortality statistics by death pathologies that relate to pharyngeal function. The authors distinguish two such pathologies, which they characterize as (typically) early deaths associated with extensive bacterial infection and a swollen pharynx, dubbed P-deaths, and (typically) later deaths associated with an atrophied pharyngeal tissue that has some bacterial inclusions but no rampant infection, dubbed p-deaths. The authors characterize these pathologies microscopically and probe their causal relation to death by intervention on the food source (dead vs live bacteria) and by mutations affecting pharyngeal function, such as *eat-2* and *phm-2*. They propose a narrative in which high pharyngeal activity exposes young worms to the risk of mechanical damage of the pharyngeal cuticular tissue leading to bacterial invasion and subsequent infection. Since cuticular damage can be repaired not all bacterial invasions end up in P-death. Repair seems to diminish P-risk and the decrease in pumping rate with age presumably decreases further the risk of damage, making P-deaths typically early events. The authors emphasize that the distinct risk of death profiles associated with these pharyngeal events contribute to explaining the variance of lifespan and deceleration of mortality specifically in an age interval around day 11, which the authors observed for wildtype at 20C.

I find the intent of the manuscript to be quite interesting and potentially suitable for Nature Communications. However, I have some confusions whose clarification would increase my ability to assess the soundness of the arguments.

Main issue

1.

There is only one main issue: is the change in the wildtype risk slope a stable reproducible phenomenon?

Let me clarify from the outset that the central idea of defining a type of death (say P-death) and using it to "deconvolve" all-cause mortality is a good one. However, I believe its fruitfulness does not at all depend on whether there is or is not a change in the risk slope around day 11. The key part is the biological characterization of a type of death that enables the stratification of mortality data. I'm no expert, but in my opinion the authors succeed in that characterization. Yet, if the authors emphasize the change of risk slope around day 11 as a key aspect of their paper, then they should be more convincing about the phenomenon.

I have not seen a deceleration of this kind at this age in lifespan data collected from about thousand wildtype worms per replicate and several replicates at 20C; eg Stroustrup et al, doi: 10.1038/nature16550. (This is an automated measurement methodology that is comparable but different in detail from the manual method and uses FUDR to avoid progeny). If I correctly read the tables, the data in Fig 1H (black curve) stem from two replicates with a combined 179 individuals. Given the centrality of the phenomenon in the paper's narrative, the authors should present multiple replicates with sufficient power and with confidence intervals, not only data pooled across replicates. This would allow one to better assess the stability of the phenomenon. As the authors know, mortality data are very noisy early in a lifespan experiment (when death events are still scarce) and late in a lifespan experiment (when the surviving population is small).

The authors re-analyze their own lifespan data from more than a decade ago by also pooling several clusters of experiments, each cluster separated by several years (UMC and UCL), one of which is corrected by a 3-day shift. This is a lot of pooling and processing across varying experimental conditions. Indeed, the change in slope of the archival analysis appears to be much

smaller than in Fig 3H.

2.

Part of my interest in this work derives from how it might help in shedding light on the temporal scaling of mortality statistics (Stroustrup et al, doi:10.1038/nature16550) . An intervention that only rescales mortality implies that all risks of death must be rescaled in time, and they must be rescaled equally.

Specifically, whether the carbenicillin experiment changes risk slopes or merely shifts them (which would be a scaling) is hard to tell by eye from Fig 2F. Carbenicillin does not appear to eliminate bumps in the risk curve, they just seem to occur later. The survival curves of wildtype on life bacteria and on carbenicillin-treated bacteria seem to be rescaled versions of one another (Fig 3D), but it's hard to tell without an analysis of accelerated failure time (AFT) residuals. The switch experiments with carbenicillin appear--again by eye--to be compatible with shifts in the lifespan distribution, which would in turn be consistent with a scaling relation between wildtype without and with carbenicillin-treated bacteria (doi:10.1038/nature16550).

The authors show convincingly that the switch from live to dead bacteria impacts P-death (bacterial infection), but it may do so to the same extent as it impacts any other risk of death. This would be interesting in its own right, but would require a scaling analysis. It would also be consistent with the fact that the authors did not find a complete abrogation of P-death upon eat-2 intervention. In fact, the eat-2 case is one of the interventions that break scaling relative to wildtype. In conjunction with the carbenicillin data, this would suggest that the scale-breaking by eat-2 might be due to whatever risks it impacts /other/ than P-death. This, too, would be interesting.

I do not wish to prescribe what connections authors should entertain. I simply reiterate what I stated at the outset: the authors' work is interesting regardless of whether the wildtype survival curve is biphasic, but if they claim it is biphasic they need to show it more convincingly by comparing replicates and by testing that the observation does not arise by chance.

Minor issues

1. Scoring pathologies

It is unclear from the text whether the scoring of pathology is quantitative to the point where a score of 4 is really twice as good as a score of 2.

2. Mortality rates

2.1

Fig 1H and other graphs showing mortality rates are really lin/lin plots displaying lin/log data; the ordinate is linear but the variable (mortality rate) is a logarithmic value and what is plotted is the exponent of some base. It would be useful to label the axis log mortality rate; it would be even more useful to show it as a mortality rate with a log axis and positive values.

2.2

There appears to be a switch in the definition of mortality rate from Fig1 H to, say, Fig 4A (and pertinent supplementary figures). What is the denominator? When calculating risks (mortality) or fraction of population that has not yet experienced the event (survival) for events that are not bound to occur, such as P-death we can go two routes: (i) unconditional survival: survival does not go to zero because there is a fraction of the population that does not experience P-death or (ii) condition the analysis on the event actually occurring; in that case the conditional survival S' relates to the unconditional survival S like $S'(t) = [S(t) - S(\infty)] / [1 - S(\infty)]$. $S'(t)$ goes

to zero, $S(t)$ does not. It seems the second definition is used in Fig 1E and Fig 1H, but the first in Fig 4A. Is that correct? It would be useful to state explicitly how P-mortality is defined.

3. General framing, conceptual

It appears that P-deaths and p-deaths add up to 100% (e.g Fig 2E / controls) and the text seems to suggest such a reading. If the authors believe there are no other pathologies at this level of observational resolution, then they really identify two classes, otherwise they identify one binary death pathology: P-deaths and "the rest" (non-P deaths). Is this really about P and p or is it more about P and non-P?

There is a somewhat strong suggestion that demographic data "contain no biological information". I am no demographer, but I would disagree with the strength of that statement. While demographic data do not settle cell-biological or biochemical mechanisms, the theme of mortality deceleration (especially late life mortality deceleration) has been treated at length by demographers. It is well known that mortality deceleration can occur not only by "frailty", which is defined in this context as phenotypic heterogeneity of a population at age zero. It can also occur if there was complete phenotypic homogeneity at age zero. Mortality deceleration is a general consequence of random processes that attain a quasi-stationary distribution over live states before they get absorbed into a dead state. As an example, consider a random walk. Let 1000 identical walkers start at exactly the same place on a line; bias the random walk towards zero (aging) and measure the rate at which the walkers cross zero. This rate conditioned on the number of walkers still alive is the mortality rate and it will decelerate. (For example see Weitz, J. S. & Fraser, H. B. Explaining mortality rate plateaus. PNAS 98, 15383-15386, 2001). Mortality deceleration can be purely a consequence of heterogeneity that is not initially present but builds up intrinsically over time due to the randomness of individual life histories. There is a voluminous literature in this regard that should be cited.

Walter Fontana

Reviewer #2 (Remarks to the Author):

Zhao et al. observe in their study "Deconvolving mortality: Two forms of death in ageing *Caenorhabditis elegans*" two distinct alterations of pharyngeal morphology in ageing worms that they define as swollen P and atrophied p morphology. The authors determine that pharyngeal swelling could be prevented by inactivation of the *E. coli* the worms are ingesting. The authors then assess the populations of p versus P-related deaths in genetic mutants and determine pumping deficient *eat-2* mutants do not suffer from P death while germline deficient *glp-1* mutants have no alterations in P deaths but instead a delay in p deaths. The authors attribute death to the pharyngeal morphological changes, however, establishing causation would require demonstrating that specific abrogation of pharyngeal atrophy would extend lifespan. Numerous previous studies have established a range of disintegrating tissue parameters that have predictive power for the individual life expectancy as demonstrated initially by the Driscoll lab (Herndon et al. 2002) and recently "health parameters" were shown to be differentially affected in a diverse set of mutants with extended lifespan (Bansal et al. 2015). Leiser et al. suggested in a most recent paper that loss of vulva integrity would mark the worm's healthspan. It is unclear in how far the current study significantly adds to the 2015 Bansal and the 2016 Leiser papers. The significance of pharyngeal dysfunction in relation to the degeneration of other tissues remains to be ascertained. Mechanistically, the authors demonstrate that proliferation of *E. coli* causes pharyngeal swelling, which could lead to death. The partitioning of the *eat-2* and *phm-2* into a DR component and an potentially "infection-related" component is interesting given the wide usage of *eat-2* mutants in longevity studies. The major limitation of the study is its phenomenological nature of characterizing the death process of nematode worms without providing new mechanistic insight.

Reviewer #3 (Remarks to the Author):

In this paper the authors perform necropsy analysis of worms who died from old age and identify a difference in the phenotype in the pharynx at the time of death. The authors identify two types of pharynxes, swollen (P death) and atrophied (p death) and show how these different phenotypes separate two different lifespan trajectories within a population. They show that the frequency of the P death is reduced by genetic (*gpl-1*) or environmental interventions (carbenicillin) and that these two type of different death explain the various shapes of lifespan curves. Authors conclude that the P death is due to bacterial infections able to colonize the pharyngeal tissue which is damaged and not properly healed due to the mechanical stress produced by pumping.

Strong points:

- Performing necropsy is an unusual and original new perspective to analyze lifespan experiments. We congratulate them to their insights.
- This is the first paper that provides an explanation on how the shapes of lifespan curves changes.
- They show that different necropsy phenotypes translate to different mortality trajectories and clearly separate two distinct groups within an isogenic population
- They provide clear evidence that eliminating P death by Carbenicillin changes the frequency of P-death and thus the clearly link it to bacterial infection.
- Their model provides an elegant and simple explanation to the changes in *gpl-1* lifespan curves that we all observed but never explained.

Major Criticisms

- The authors do a good job of showing that reducing big P extends lifespan. The one experiment we would like to see is some form of a "pharyngeal stress" that increases the occurrence of big P relative to small p and thus shifts the mortality curve. This could be achieved by either increasing the pumping rate by serotonin, adding some beads to the food that hurt the pharynx, by using strains that have defects in their innate immunity, mutants that have wound healing defects or by using a more pathogenic strain. Any experimental intervention that increases the fraction of P death will do. We think this to be crucial as P-death is the form of death the authors define and characterize.

Minor Criticisms

- From what we understand from the data, but what is not so clear in the text the authors have delineated one form of death, which is P death. The other form of deaths the call p death seems to be a catch all for probably a number of other causes of death.
 - Please include measurements of young pharynxes (L4 or day 1 or day 2) into the graph 1D, as to support the model in the supplementary data that shows small p to be due to attrition.
- The authors claim to present necropsy analysis of the animals at the time of death. Is there any correlation of the P death with other phenotypes at the time of death? Do the animals look "older" in their body or present any other signs of infection at the time of death who demonstrate decay of the body? Or it is just a swollen pharynx. How those necropsies of P death compare with p death? Please provide a series of P and p death images that show the entire animals for the reader to appreciate the differences.
- The mortality curve shown for P-death in Fig 1H looks very distinct from the theoretical mortality shown in Fig 4A. We appreciate the difficulty of getting a P-death curve beyond day 12, as there will be very P-deaths beyond this age. However given the importance such an experimentally verified curve should be provided. It may be possible to obtain and estimate for this curve by subtracting the carbenicillin mortality from the control mortality (Fig 1F) which should result in the P-death mortality.
- In figure 2A the authors try to describe the difference in bacteria distribution and the possible correlation with the infection and the phenotype. They quantify the distribution of the bacteria in p death and P death in figure 2B. The image of p-death show a good amount of bacteria widespread

throughout the pharynx which makes it hard to appreciate the differences except in the grinder. Authors need to use more representative pictures of what they are trying to show.

- Please provide a better explanation on how the models in figure 4 were constructed. Were these simulations, were there any equations, etc.

Authors' responses to reviewers' comments

Responses are in blue font.

Reviewer #1 (Remarks to the Author):

The manuscript by Zhao et al aims at stratifying *C. elegans* mortality statistics by death pathologies that relate to pharyngeal function. The authors distinguish two such pathologies, which they characterize as (typically) early deaths associated with extensive bacterial infection and a swollen pharynx, dubbed P-deaths, and (typically) later deaths associated with an atrophied pharyngeal tissue that has some bacterial inclusions but no rampant infection, dubbed p-deaths. The authors characterize these pathologies microscopically and probe their causal relation to death by intervention on the food source (dead vs live bacteria) and by mutations affecting pharyngeal function, such as *eat-2* and *phm-2*. They propose a narrative in which high pharyngeal activity exposes young worms to the risk of mechanical damage of the pharyngeal cuticular tissue leading to bacterial invasion and subsequent infection. Since cuticular damage can be repaired not all bacterial invasions end up in P-death. Repair seems to diminish P-risk and the decrease in pumping rate with age presumably decreases further the risk of damage, making P-deaths typically early events. The authors emphasize that the distinct risk of death profiles associated with these pharyngeal events contribute to explaining the variance of lifespan and deceleration of mortality specifically in an age interval around day 11, which the authors observed for wildtype at 20C.

I find the intent of the manuscript to be quite interesting and potentially suitable for Nature Communications. However, I have some confusions whose clarification would increase my ability to assess the soundness of the arguments.

Main issue

1.

There is only one main issue: is the change in the wildtype risk slope a stable reproducible phenomenon?

Let me clarify from the outset that the central idea of defining a type of death (say P-death) and using it to "deconvolve" all-cause mortality is a good one. However, I believe its fruitfulness does not at all depend on whether there is or is not a change in the risk slope around day 11. The key part is the biological characterization of a type of death that enables the stratification of mortality data. I'm no expert, but in my opinion the authors succeed in that characterization. Yet, if the authors emphasize the change of risk slope around day 11 as a key aspect of their paper, then they should be more convincing about the phenomenon.

I have not seen a deceleration of this kind at this age in lifespan data collected from about thousand wildtype worms per replicate and several replicates at 20C; eg Stroustrup et al, doi:10.1038/nature16550. (This is an automated measurement methodology that is comparable but different in detail from the manual method and uses FUDR to avoid progeny). If I correctly read the tables, the data in Fig 1H (black curve) stem from two replicates with a combined 179 individuals. Given the centrality of the phenomenon in the paper's narrative, the authors should present multiple replicates with sufficient power and with confidence intervals, not only data pooled across replicates. This would allow one to better assess the stability of the phenomenon. As the authors know, mortality data are very noisy early in a lifespan experiment (when death events are still scarce) and late in a lifespan experiment (when the surviving population is small).

The authors re-analyze their own lifespan data from more than a decade ago by also pooling several clusters of experiments, each cluster separated by several years (UMC and UCL), one of which is corrected by a 3-day shift. This is a lot of pooling and processing across varying experimental conditions. Indeed, the change in slope of the archival analysis appears to be much smaller than in Fig 3H.

Authors' response: Several important points are raised here.

The mid-life deceleration in mortality rate of *C. elegans* maintained in the presence of proliferating *E. coli* has been described in a number of studies, mainly published in the 1990s; see e.g. Brooks et al. 1994, Vaupel et al. 1994, and Johnson et al. 2001 (cited in the manuscript). Thus, it was not our aim to establish the existence or the biological relevance of this phenomenon but, rather, to provide an explanation for why it occurs, using new approaches.

We agree that the sample size used to generate the mortality profiles in our original figure (Fig 1G) was rather small. We have now combined data from eleven trials, thereby increasing the sample size from 179 to 622 animals, and provided statistical analysis of the slope change, which is statistically significant, particularly on days 10 and 11 (new Fig. 1G).

To rule out effects of the variation among data from multiple trials, we plotted the survival of each trial to show that the lifespan is highly reproducible (Supplementary Fig. 3), and conducted an additional single large trial with 585 animals, measuring mortality rate and performing mortality deconvolution (Supplementary Fig. 3). This also showed a surge of mortality in mid-life.

About the lack of mid-life mortality deceleration in the data collected using the lifespan machine: this is expected in most trials, where antibiotics and UV-irradiation were used, which suppress P death (Fig. 2E). For trials where live bacteria were used: we tested the *E. coli* strain used in the lifespan machine, NEC937B, and showed that it significantly reduced the frequency of P death (Supplementary Fig. 7), which is predicted to reduce the early surge in mortality. We also examined data from another recently developed automated lifespan apparatus, the worm corral of Zachary Pincus, Washington University at St Louis (Zhang et al, 2016). From analysis of worm images captured by the worm corral system, we established that P death is absent from worms maintained in this system (the exact cause for this is unclear, but there are several possibilities). As expected from this, no early-life deceleration was observed in worm corral mortality data (*spe-9* sterile mutants). These new findings have been added to the paper.

2.

Part of my interest in this work derives from how it might help in shedding light on the temporal scaling of mortality statistics (Stroustrup et al, doi:10.1038/nature16550) . An intervention that only rescales mortality implies that all risks of death must be rescaled in time, and they must be rescaled equally.

Specifically, whether the carbenicillin experiment changes risk slopes or merely shifts them (which would be a scaling) is hard to tell by eye from Fig 2F. Carbenicillin does not appear to eliminate bumps in the risk curve, they just seem to occur later. The survival curves of wildtype on live bacteria and on carbenicillin-treated bacteria seem to be rescaled versions of one another (Fig 3D), but it's hard to tell without an analysis of accelerated failure time (AFT) residuals. The switch experiments with carbenicillin appear--again by eye--to be compatible with shifts in the lifespan distribution, which would in turn be consistent with a scaling relation between wildtype without and with carbenicillin-treated bacteria (doi:10.1038/nature16550).

The authors show convincingly that the switch from live to dead bacteria impacts P-death (bacterial infection), but it may do so to the same extent as it impacts any other risk of death. This would be interesting in its own right, but would require a scaling analysis. It would also be consistent with the fact that the authors did not find a complete abrogation of P-death upon eat-2 intervention. In fact, the eat-2 case is one of the interventions that break scaling relative to wildtype. In conjunction with the carbenicillin data, this would suggest that the scale-breaking by

eat-2 might be due to whatever risks it impacts /other/ than P-death. This, too, would be interesting.

I do not wish to prescribe what connections authors should entertain. I simply reiterate what I stated at the outset: the authors' work is interesting regardless of whether the wildtype survival curve is biphasic, but if they claim it is biphasic they need to show it more convincingly by comparing replicates and by testing that the observation does not arise by chance.

Authors' response: The question of how interventions that differentially affect P and p subpopulations may relate to the observations on temporal scaling made by Stroustrup et al is a very interesting one. Responding to the reviewer, we did some analysis using AFT modelling in combination with mortality deconvolution. Preliminary analysis of our data showed similar scaling of different p-only worm populations (e.g. p deaths from populations on live *E. coli* compared to all worms maintained on carbenicillin), but not with populations with both P and p deaths. Our findings raise the possibility that in conditions permissive for P death, the presence of the P subpopulation may mask the scaling of mortality patterns in p subpopulations. In that case, the absence of P deaths under culture conditions used in the lifespan machine data was surely fortunate, since it allowed the detection of scaling effects that P death sub-populations might otherwise likely have obscured.

Our scaling analysis is still very preliminary, nonetheless it suggests to us that combining the power of scaling analysis as described by Stroustrup et al with that of P vs p mortality deconvolution as described here could significantly enhance the power of biodemography to interrogate ageing biology in *C. elegans*. If possible we would like to conduct further investigations here, hopefully collaboratively. We therefore feel that proper characterization of the scaling of p death subpopulations is beyond the scope of this study, but would be happy to share the preliminary results.

Minor issues

1. Scoring pathologies

It is unclear from the text whether the scoring of pathology is quantitative to the point where a score of 4 is really twice as good as a score of 2.

Authors' response: The pathology scores for pharynxes, distal gonads and uterine tumours are on an ordinal scale which run from 1 to 5, where 1 is the score of a day 1 wild-type adult. A score of 4 indicates a pathology that is much more severe than a 2, but is not twice as severe. Severity of yolk pools and intestinal atrophy was estimated more accurately, by measuring the size of the yolk pools and intestine relative to body size. We have expanded the description of the scoring system in the methods section.

2. Mortality rates

2.1

Fig 1H and other graphs showing mortality rates are really lin/lin plots displaying lin/log data; the ordinate is linear but the variable (mortality rate) is a logarithmic value and what is plotted is the exponent of some base. It would be useful to label the axis log mortality rate; it would be even more useful to show it as a mortality rate with a log axis and positive values.

Authors' response: The axis has been labeled "log mortality rate", as suggested.

2.2

There appears to be a switch in the definition of mortality rate from Fig1 H to, say, Fig 4A (and pertinent supplementary figures). What is the denominator? When calculating risks (mortality) or

fraction of population that has not yet experienced the event (survival) for events that are not bound to occur, such as P-death we can go two routes: (i) unconditional survival: survival does not go to zero because there is a fraction of the population that does not experience P-death or (ii) condition the analysis on the event actually occurring; in that case the conditional survival S' relates to the unconditional survival S like $S'(t) = [S(t) - S(\infty)] / [1 - S(\infty)]$. $S'(t)$ goes to zero, $S(t)$ does not. It seems the second definition is used in Fig 1E and Fig 1H, but the first in Fig 4A. Is that correct? It would be useful to state explicitly how P-mortality is defined.

Authors' response: We have changed Fig 1H to make the definition of mortality rate consistent, and made the definition clear in the Methods section. All the mortality rates are now calculated as unconditional survival, i.e. censoring the other type of death, as this will help clearly illustrate the changes in P death frequency in Fig 4.

3. General framing, conceptual

It appears that P-deaths and p-deaths add up to 100% (e.g Fig 2E / controls) and the text seems to suggest such a reading. If the authors believe there are no other pathologies at this level of observational resolution, then they really identify two classes, otherwise they identify one binary death pathology: P-deaths and "the rest" (non-P deaths). Is this really about P and p or is it more about P and non-P?

Authors' response: Necropsy data shows that all the worms die with either P or p. Our designation P or p deaths merely denotes this, not that the worms have died as a consequence of an enlarged or shrunken pharynx. However, in the case of P, the swelling is indicative of a fatal bacterial infection; in the case of p, the atrophy immediately precedes death, but it is not clear that it contributes to mortality.

There is a somewhat strong suggestion that demographic data "contain no biological information". I am no demographer, but I would disagree with the strength of that statement. While demographic data do not settle cell-biological or biochemical mechanisms, the theme of mortality deceleration (especially late life mortality deceleration) has been treated at length by demographers. It is well known that mortality deceleration can occur not only by "frailty", which is defined in this context as phenotypic heterogeneity of a population at age zero. It can also occur if there was complete phenotypic homogeneity at age zero. Mortality deceleration is a general consequence of random processes that attain a quasi-stationary distribution over live states before they get absorbed into a dead state. As an example, consider a random walk. Let 1000 identical walkers start at exactly the same place on a line; bias the random walk towards zero (aging) and measure the rate at which the walkers cross zero. This rate conditioned on the number of walkers still alive is the mortality rate and it will decelerate. (For example see Weitz, J. S. & Fraser, H. B. Explaining mortality rate plateaus. PNAS 98, 15383-15386, 2001). Mortality deceleration can be purely a consequence of heterogeneity that is not initially present but builds up intrinsically over time due to the randomness of individual life histories. There is a voluminous literature in this regard that should be cited.

Authors' response: We agree that our original description of lifespan being "a demographic parameter devoid of biological information" is overstated. To make clearer what was meant by this passage we have rephrased it as follows (including the context of the statement): "Although much progress has been made in terms of identifying genes and pathways that affect lifespan, the underlying mechanisms of ageing remain poorly defined. One obstacle has been the difficulty of relating gene function to lifespan, given that the latter is a numeric, demographic parameter which contains little information about biological processes or structures to which gene function can readily be related. [...] Identification of lethal senescent pathologies may provide us with the missing link between the biochemical function of gene products and their effects on lifespan."

Reviewer #2 (Remarks to the Author):

Zhao et al. observe in their study “Deconvolving mortality: Two forms of death in ageing *Caenorhabditis elegans*” two distinct alterations of pharyngeal morphology in ageing worms that they define as swollen P and atrophied p morphology. The authors determine that pharyngeal swelling could be prevented by inactivation of the *E. coli* the worms are ingesting. The authors then assess the populations of p versus P-related deaths in genetic mutants and determine pumping deficient *eat-2* mutants do not suffer from P death while germline deficient *glp-1* mutants have no alterations in P deaths but instead a delay in p deaths. The authors attribute death to the pharyngeal morphological changes, however, establishing causation would require demonstrating that specific abrogation of pharyngeal atrophy would extend lifespan.

Authors' response: A contribution of pharyngeal senescent atrophy to mortality is an interesting possibility that we do not address in the paper, which largely focuses on P death. If the mechanisms leading to senescent atrophy could be identified, then this approach could be considered.

Numerous previous studies have established a range of disintegrating tissue parameters that have predictive power for the individual life expectancy as demonstrated initially by the Driscoll lab (Herndon et al. 2002) and recently “health parameters” were shown to be differentially affected in a diverse set of mutants with extended lifespan (Bansal et al. 2015). Leiser et al. suggested in a most recent paper that loss of vulva integrity would mark the worm’s healthspan. It is unclear in how far the current study significantly adds to the 2015 Bansal and the 2016 Leiser papers.

Authors' response: The reviewer is correct that many studies have shown that interventions that extend *C. elegans* lifespan can slow the decline in various behavioural markers of health and delay the appearance of senescent pathologies. A number of these earlier studies (e.g. Garigan et al. 2002; Herndon et al. 2002) provided a vital jumping off point for our study, which presents a number of novel findings, including the following. (i) A subset of *C. elegans* die early as the result of bacterial infection of the pharynx. (ii) This appears to be due to activity dependent mechanical damage to the pharyngeal cuticle. (iii) *eat-2* mutant longevity is apparently attributable to suppression of P. (iv) Combination of necropsy and mortality data allows mortality deconvolution, a novel approach which renders intelligible the complex shapes of survival curves and mortality profiles.

The significance of pharyngeal dysfunction in relation to the degeneration of other tissues remains to be ascertained.

Authors' response: This is an interesting point not addressed in the manuscript. In preliminary studies of elderly wild-type hermaphrodites, we have sometimes observed animals with a swollen pharynx and bacterial infection contained within it, that still show vigorous movement; however, when the infection has spread from the pharynx to the body cavity the animals are either dead or close to death. This and other studies suggest that invasion of the body cavity by *E. coli* leads rapidly to death. However, these studies are incomplete, and we feel that they are beyond the scope of the present study.

Mechanistically, the authors demonstrate that proliferation of *E. coli* causes pharyngeal swelling, which could lead to death. The partitioning of the *eat-2* and *phm-2* into a DR component and an potentially "infection-related" component is interesting given the wide usage of *eat-2* mutants in longevity studies.

Authors' response: We are glad that the reviewer finds this point interesting. We have now extended our examination of the effects on P frequency of reducing pumping rate by looking at a range of additional pumping-defective mutants, including *eat-5*, *eat-13* and *phm-3*. The majority of these mutants showed reductions in frequency of P (Fig. 3C). This further confirms the effects of wild-type high rates of pharyngeal pumping on P. The new tests also included verification of the effects of *phm-2(ad597)* and *eat-2(ad1116)*. These confirmed the effects of *phm-2* on P, but led to a minor amendment of the *eat-2* results: that rather than abrogating P entirely, *eat-2(ad1116)* causes a significant reduction in P frequency, down to <20%. This minor change does not change the conclusion that suppression of P contributes to *eat-2* longevity (Fig. 3D). Our findings, using standard culture conditions, imply that *eat-2* longevity is almost entirely attributable to suppression of P.

However from our own experience with *eat-2*, and discussions with folks from other labs who have worked with *eat-2*, it appears that *eat-2* longevity is subject to variability in subtle differences in standard lab culture conditions that remain poorly understood. It therefore seems possible that under different standard culture conditions, *eat-2* might increase lifespan in the p sub-population (like *phm-2* does). This is why we have not made more of a fuss about these findings about *eat-2*.

The major limitation of the study is its phenomenological nature of characterizing the death process of nematode worms without providing new mechanistic insight.

Authors' response: It is true that some of the findings in this study are descriptive (phenomenological), for example, the existence of two types of death in *C. elegans*. But descriptive findings can sometimes be interesting and important (for example, the recent discovery of a potentially Earth-like planet circling our neighbouring star, Proxima Centauri). The existence of two forms of death in *C. elegans* does provide new insight into population level analysis; for example, explaining the long-standing mystery of the mortality rate deceleration in *C. elegans*, first observed some 25 years ago. In terms of actual biological mechanism, the study does present an account of the mechanisms leading to P death, that involves activity-dependent mechanical senescence, and an interplay between innate immunity and infection; so there is at least some new mechanistic insight. It also reveals a new mechanism that contributes to *eat* mutant longevity. But most importantly, it presents an approach that we believe will help understand the mechanisms by which genes specify ageing and lifespan by taking into consideration the biology that lies between them: that of senescent pathology.

Reviewer #3 (Remarks to the Author):

In this paper the authors perform necropsy analysis of worms who died from old age and identify a difference in the phenotype in the pharynx at the time of death. The authors identify two types of pharynxes, swollen (P death) and atrophied (p death) and show how these different phenotypes separate two different lifespan trajectories within a population. They show that the frequency of the P death is reduced by genetic (*gpl-1*) or environmental interventions (carbenicillin) and that these two type of different death explain the various shapes of lifespan curves. Authors conclude that the P death is due to bacterial infections able to colonize the pharyngeal tissue which is damaged and not properly healed due to the mechanical stress produced by pumping.

Strong points:

- Performing necropsy is an unusual and original new perspective to analyze lifespan experiments. We congratulate them to their insights.
- This is the first paper that provides an explanation on how the shapes of lifespan curves changes.
- They show that different necropsy phenotypes translate to different mortality trajectories and clearly separate two distinct groups within an isogenic population
- They provide clear evidence that eliminating P death by Carbenicillin changes the frequency of P-death and thus the clearly link it to bacterial infection.

-Their model provides an elegant and simple explanation to the changes in glp-1 lifespan curves that we all observed but never explained.

Major Criticisms

- The authors do a good job of showing that reducing big P extends lifespan. The one experiment we would like to see is some form of a “pharyngeal stress” that increases the occurrence of big P relative to small p and thus shifts the mortality curve. This could be achieved by either increasing the pumping rate by serotonin, adding some beads to the food that hurt the pharynx, by using strains that have defects in their innate immunity, mutants that have wound healing defects or by using a more pathogenic strain. Any experimental intervention that increases the fraction of P death will do. We think this to be crucial as P-death is the form of death the authors define and characterize.

Authors' response: Thinking along similar lines, we previously attempted to increase mechanical stress to the grinder by feeding the worms with *E. coli* mixed with finely ground glass but, unfortunately, the results were inconclusive. At the reviewer's suggestion we also exposed worms to serotonin, which increased the pumping rate when worms were off the bacterial lawn, where they normally pump less, but not when the worms were in the lawn. However, extended exposure to high levels of serotonin appeared to have a toxic effect on the worms and caused premature death, and did not increase P death.

We also took a different approach to increase pharyngeal stress using a *ced-1* mutant previously shown to be hyper-susceptible to pharyngeal invasion by pathogenic bacteria. This shortened the lifespan of worms raised on proliferating *E. coli*, and mortality deconvolution showed an increase in the frequency (but not timing) of P death, as well as accelerated mortality in the p sub-population (new Fig. 3G-H). We thank the reviewer for these constructive suggestions.

Minor Criticisms

- From what we understand from the data, but what is not so clear in the text the authors have delineated one form of death, which is P death. The other form of deaths the call p death seems to be a catch all for probably a number of other causes of death.

Authors' response: See response to referee 1, point 3.

- Please include measurements of young pharynxes (L4 or day 1 or day 2) into the graph 1D, as to support the model in the supplementary data that shows small p to be due to attrition.

Authors' response: The individual plots in Fig. 1E shows that the pharynx initially grows, reaching maximum size around day 7-10, and then as the p worms grow old, it shrinks. Therefore, measurement of 10-day old pharynxes has been added to Fig. 1C (originally Fig. 1D).

- The authors claim to present necropsy analysis of the animals at the time of death. Is there any correlation of the P death with other phenotypes at the time of death? Do the animals look “older” in their body or present any other signs of infection at the time of death who demonstrate decay of the body? Or it is just a swollen pharynx. How those necropsies of P death compare with p death? Please provide a series of P and p death images that show the entire animals for the reader to appreciate the differences.

Authors' response: We have added to the supplement a series of P and p death images showing the whole animal, as suggested (Supplementary Fig. 2). This is an interesting question which we are currently investigating. Preliminary evidence suggests that P and p death animals do differ with respect to other aspects of pathology, for example patterns of *E. coli* infection. The issue of other pathologies in P and p worms is raised by referee 2 (see corresponding response). However, we believe that a detailed analysis on other pathologies is beyond the scope of this study.

•The mortality curve shown for P-death in Fig 1H looks very distinct from the theoretical mortality shown in Fig 4A. We appreciate the difficulty of getting a P-death curve beyond day 12, as there will be very P-deaths beyond this age. However given the importance such an experimentally verified curve should be provided. It may be possible to obtain and estimate for this curve by subtracting the carbenicillin mortality from the control mortality (Fig 1F) which should result in the P-death mortality.

Authors' response: The difference in the original figures is due to two different ways of calculating mortality rate: conditional survival in the original Fig. 1H, and unconditional in Fig. 4. We have now made the definition consistent by calculating the mortality rate using the unconditional method, i.e. censoring the other type of death (so they count towards the total number of event), as this will help illustrate the changes in P death frequency in Fig 4.

•In figure 2A the authors try to describe the difference in bacteria distribution and the possible correlation with the infection and the phenotype. They quantify the distribution of the bacteria in p death and P death in figure 2B. The image of p-death show a good amount of bacteria widespread throughout the pharynx which makes it hard to appreciate the differences except in the grinder. Authors need to use more representative pictures of what they are trying to show.

Authors' response: We have provided images that are more representative of the p death phenotype described.

•Please provide a better explanation on how the models in figure 4 were constructed. Were these simulations, were there any equations, etc.

Authors' response: A more detailed explanation has been added to the figure legend. Briefly, the “original” P and p survival curves were adapted from the P and p death survival calculations in Fig. 1G, and the survival proportions of the whole population were calculated, assuming that P death is 40% of the population and p death is 60%, which is an idealized approximation to the real data. For the transformations: to alter the timing of either type of death, the survival curve of that particular type of death is shifted to the right. The resulting survival data were then used to calculate the survival curve of the whole population (Fig. 4), as well as the log mortality rate of each subpopulation (Supplementary Fig. 11). To alter the frequency of both types of death, an increased proportion of one type of death (and the concurrent reduction of the other) was used to calculate the survival of the whole population, and the resulting data were used to calculate the log mortality rates.

Reviewer #1 (Remarks to the Author)

My comments or concerns were mostly addressed and I support publication of the manuscript. This said, I would like to make a final comment in light of the additional information the authors provided. It is but a comment and I leave it to the authors whether or in what way they wish to take it into account.

By my lights, the hazard data can be summarized as follows (in no particular order): (i) The mortality rates of the n=585 experiment in the supplement appear to be very noisy where it matters the most. (ii) The location of the mortality rate change of interest appears to occur quantitatively at different ages (though always early) across the literature cited. (iii) According to the authors, the data from co-author Zach Pincus' approach in the context of automatic lifespan acquisition do not evidence the biphasic switch. (iv) Our data using OP50 with a deletion of *uvrA*, Figure 1h of doi: 10.1038/nature16550, does not show the effect either (black curve; the green curve is with UV-irradiated bacteria as food source, and Fig 1i is the superposition of both curves in residual time after removal of the scale factor). (v) Tests by the authors with the strain we used appear to confirm our result. (vi) Additional replicate trials by the authors evidence the biphasic switch.

What are we to make of this? I have no difficulty believing that there is an effect, but it seems rather finicky in its dependence on experimental conditions, which may or may not be limited to the particular bacterial strain used as food source. Conceptually, my take from this is as follows.

Medawar (in his now classic Inaugural Lecture at the University College London, 1951) and many after him, perhaps most notably Strehler and Mildvan (Science, 132/3418, pp. 14-21, 1960), made a distinction between aging as an intrinsic decrease in the capacity of restoring a viable physiological state after a challenge and the challenges themselves (that is, the events proximally causing death). In the Strehler-Mildvan idealization, a challenge whose magnitude exceeds that capacity leads (instantly) to death. For the sake of argument, we might want to wrap together the intrinsic decline in resilience with the challenges that arise from within the organism (perhaps as a result of that very decline) as intrinsic risk and distinguish between it and the extrinsic risk due to challenges originating in the environment. Both contribute to mortality, but it is the intrinsic stuff that we have in mind when we refer to aging. When measuring aging through mortality, we then deal, perhaps inevitably, with some degree of confounding between intrinsic and extrinsic risks of death. The moment a daredevil jumps from a cliff with a squirrel suit her risk of death goes through the roof, but that doesn't mean that she has suddenly aged.

Some bacterial food strains may be more toxic than others in the sense that they may be more infectious. The UV-irradiated bacteria (which at the very minimum do not reproduce) are not infectious, so no P-death. The OP50 delta *uvrA* strain that we used is perhaps less infectious because of the deletion in a DNA damage repair component. The strain the authors used (OP50 plus RFP plasmid) seems more infectious. As the authors note, individuals who pump a lot (and there is a distribution of pumping activity in an isogenic wildtype population) may damage their pharynx more than others. But a damaged pharynx doesn't kill; it may, however, increase the risk of death systemically and it also increases permissiveness for many potential extrinsic causes of death. Now, if the population is in an environment where this particular intrinsic risk component (a damaged pharynx) sufficiently lowers the threshold for bacterial infection (an extrinsic risk), the population will suddenly be more sensitive to the infectiousness of the environment, and the environment will now "project out" that subpopulation which is most at risk to infection.

Stated in the language of scaling: if the population is in an environment whose challenges do not dominate the intrinsic lifespan, scaling might be a signature of intrinsic aging (in *C. elegans*); in an environment whose challenges begin to effectively stratify the population (because some extrinsic risks start to dominate lifespan), multiphasic risk profiles might suddenly emerge and scaling breaks. (Not because the mechanism of aging changes, but because the environment emphasizes

certain intrinsic risks more than others.) Since environment and genetics can, to some extent, mimic each other in terms of phenotypic outcomes, we might observe the same effect by keeping the environment constant while changing the genetic makeup.

I'm sure that I'm not saying much that the authors don't already know. They title their paper "deconvolution of mortality" and not "deconvolution of aging". Yet, in the main text, it sounds as if they want to deconvolve aging. Indeed, the major message of the paper is that worm autopsies, if quantitative and aggregated, provide a bridge from demography (mortality statistics) to mechanistic physiology that will yield insights into the fundamental mechanisms of aging. I could not agree more. I'm trying to suggest that if the authors were to discuss the above issue---i.e. how the signature of aging is buried in mortality---more fully in the light of the "instability" of the phenomenon of biphasic hazard, they would strengthen, not weaken, the appeal of their vision.

One final administrative remark. The appropriate citation of our work in this context is not the Nature Methods paper but the 2016 Nature paper on scaling, as it includes the NEC937 live vs UV-irradiated comparison in its Fig. 1. I would like to ask the authors to swap the current citation with the proposed one.

Walter Fontana

Reviewer #2 (Remarks to the Author)

Zhou et al. have decided to refrain from addressing any of the comments asking for some mechanistic insight. It seems the group is more interested in observational studies as evidenced by their misplaced comment regarding the discovery of an Earth-like planet. The study is technically fine and provides insight into distinct death-associated pharyngeal phenotypes that are important to distinguish the longevity phenotypes of eat-2 mutants that have defects in pharyngeal pumping and glp-1 mutants that display longevity due to germline defects. While the concept of death due to bacterial infection or to tissue atrophy is nothing new, the study makes a good addition to previous reports, as pointed out in my initial comments, on the necropsy of laboratory *C. elegans*, which will be useful for the community.

Reviewer #3 (Remarks to the Author)

In this manuscript the authors de-convolute different causes of mortality and show that some of the more puzzling aspects of lifespan data and the resulting lifespan curves can be explained by separating the population by their mode of death. This is a very original insight about a phenomenon we all have observed but were not able to explain. The data are solid and the authors have increased the numbers provided with the last version.

In the initial manuscript I had only one major concern, which was the testing of a strain/condition with increased stress or susceptibility to infection in the pharynx. By testing ced-1 mutants, the authors have provided an excellent example.

In my view, there is additional insight that the paper provides, which is of importance but is currently hidden. By analyzing P and p deaths for animals raised in different environmental conditions (automation, NGM) , showing that the frequency of P death is environment dependent, the authors also provide an explanation on why some lifespan extension mechanisms only extend lifespan under some, but not other conditions. If a lifespan mechanism prevents a form of death that does not occur in a given environment, it will not increase lifespan.

I am in full support publishing the manuscript

Authors' responses to reviewers' comments

Responses are in blue font.

Reviewer #1 (Remarks to the Author):

My comments or concerns were mostly addressed and I support publication of the manuscript. This said, I would like to make a final comment in light of the additional information the authors provided. It is but a comment and I leave it to the authors whether or in what way they wish to take it into account.

By my lights, the hazard data can be summarized as follows (in no particular order): (i) The mortality rates of the n=585 experiment in the supplement appear to be very noisy where it matters the most. (ii) The location of the mortality rate change of interest appears to occur quantitatively at different ages (though always early) across the literature cited. (iii) According to the authors, the data from co-author Zach Pincus' approach in the context of automatic lifespan acquisition do not evidence the biphasic switch. (iv) Our data using OP50 with a deletion of *uvrA*, Figure 1h of doi:10.1038/nature16550, does not show the effect either (black curve; the green curve is with UV-irradiated bacteria as food source, and Fig 1i is the superposition of both curves in residual time after removal of the scale factor). (v) Tests by the authors with the strain we used appear to confirm our result. (vi) Additional replicate trials by the authors evidence the biphasic switch.

What are we to make of this? I have no difficulty believing that there is an effect, but it seems rather finicky in its dependence on experimental conditions, which may or may not be limited to the particular bacterial strain used as food source. Conceptually, my take from this is as follows.

Medawar (in his now classic Inaugural Lecture at the University College London, 1951) and many after him, perhaps most notably Strehler and Mildvan (Science, 132/3418, pp. 14-21, 1960), made a distinction between aging as an intrinsic decrease in the capacity of restoring a viable physiological state after a challenge and the challenges themselves (that is, the events proximally causing death). In the Strehler-Mildvan idealization, a challenge whose magnitude exceeds that capacity leads (instantly) to death. For the sake of argument, we might want to wrap together the intrinsic decline in resilience with the challenges that arise from within the organism (perhaps as a result of that very decline) as intrinsic risk and distinguish between it and the extrinsic risk due to challenges originating in the environment. Both contribute to mortality, but it is the intrinsic stuff that we have in mind when we refer to aging. When measuring aging through mortality, we then deal, perhaps inevitably, with some degree of confounding between intrinsic and extrinsic risks of death. The moment a daredevil jumps from a cliff with a squirrel suit her risk of death goes through the roof, but that doesn't mean that she has suddenly aged.

Some bacterial food strains may be more toxic than others in the sense that they may be more infectious. The UV-irradiated bacteria (which at the very minimum do not reproduce) are not infectious, so no P-death. The OP50 delta *uvrA* strain that we used is perhaps less infectious because of the deletion in a DNA damage repair component. The strain the authors used (OP50 plus RFP plasmid) seems more infectious. As the authors note, individuals who pump a lot (and there is a distribution of pumping activity in an isogenic wildtype population) may damage their pharynx more than others. But a damaged pharynx doesn't kill; it may, however, increase the risk of death systemically and it also increases permissiveness for many potential extrinsic causes of death. Now, if the population is in an environment where this particular intrinsic risk component (a damaged pharynx) sufficiently lowers the threshold for bacterial infection (an extrinsic risk), the population will suddenly be more sensitive to the infectiousness of the environment, and the environment will now "project out" that subpopulation which is most at risk to infection.

Stated in the language of scaling: if the population is in an environment whose challenges do not dominate the intrinsic lifespan, scaling might be a signature of intrinsic aging (in *C. elegans*); in an environment whose challenges begin to effectively stratify the population (because some extrinsic risks start to dominate lifespan), multiphasic risk profiles might suddenly emerge and scaling breaks. (Not because the mechanism of aging changes, but because the environment emphasizes certain intrinsic risks more than others.) Since environment and genetics can, to some extent, mimic each other in terms of phenotypic outcomes, we might observe the same effect by keeping the environment constant while changing the genetic makeup.

I'm sure that I'm not saying much that the authors don't already know. They title their paper "deconvolution of mortality" and not "deconvolution of aging". Yet, in the main text, it sounds as if they want to deconvolve aging. Indeed, the major message of the paper is that worm autopsies, if quantitative and aggregated, provide a bridge from demography (mortality statistics) to mechanistic physiology that will yield insights into the fundamental mechanisms of aging. I could not agree more. I'm trying to suggest that if the authors were to discuss the above issue---i.e. how the signature of aging is buried in mortality---more fully in the light of the "instability" of the phenomenon of biphasic hazard, they would strengthen, not weaken, the appeal of their vision.

Authors' response: These remarks are very incisive, and the last suggestion is a very good one. We have now addressed this point in the discussion, as follows.

"[mortality deconvolution] provides a route to discover the biological mechanisms of ageing underlying survival curves and mortality profiles in this powerful model organism.

But is P death attributable to ageing, or is it merely the result of bacterial infection? In order to maximise the contribution of intrinsic determinants of senescence, would it not be better simply to exclude P death, e.g. by using antibiotics? Certainly, excluding P death can facilitate analysis of intrinsic determinants of senescence. However, our results suggest that the P sub-population is itself a useful model for studying the biology of ageing. For example, they suggest that damage incurred early (mechanical damage to the pharyngeal cuticle, and contained infections within pharyngeal tissue) lie latent during early-mid adulthood, but then recrudescence in later life, potentially due to action of intrinsic determinants of senescence (Supplementary Fig. 8e). Thus, the P sub-population may be used to investigate how early damage can interact with later intrinsic causes of ageing to determine senescent pathology and mortality."

This discussion implicitly refers to a theory paper that we have in preparation, which was inspired by the present work. This presents a new model for the relationship between early damage and later pathology. Here early damage lies latent, and then in later life is a precondition for senescent pathology. For example, in mammals somatic mutations in early adulthood can generate precancerous cells that are initially harmless, but then proliferate in later life due to senescent changes in tissue microenvironment. Similarly, mechanical damage to joints in early life can lead to osteoarthritis in later life (c.f. mechanical damage in the pharynx).

We thank the reviewer for their many useful contributions to this study.

One final administrative remark. The appropriate citation of our work in this context is not the Nature Methods paper but the 2016 Nature paper on scaling, as it includes the NEC937 live vs UV-irradiated comparison in its Fig. 1. I would like to ask the authors to swap the current citation with the proposed one.

Authors' response: We have made this swap.

Walter Fontana

Reviewer #2 (Remarks to the Author):

Zhou et al. have decided to refrain from addressing any of the comments asking for some mechanistic insight. It seems the group is more interested in observational studies as evidenced by their misplaced comment regarding the discovery of an Earth-like planet. The study is technically fine and provides insight into distinct death-associated pharyngeal phenotypes that are important to distinguish the longevity phenotypes of eat-2 mutants that have defects in pharyngeal pumping and glp-1 mutants that display longevity due to germline defects. While the concept of death due to bacterial infection or to tissue atrophy is nothing new, the study makes a good addition to previous reports, as pointed out in my initial comments, on the necropsy of laboratory *C. elegans*, which will be useful for the community.

Authors' response: We thank the reviewer for their contribution. On the subject of mechanistic insight, we agree that the new phenomena described in this study raise many new questions about mechanism, which include the following: How exactly does fast pharyngeal pumping increase P frequency? Can the putative mechanical senescence of the pharyngeal cuticle be understood in terms of forces exerted by the pharynx and biomechanical properties of the cuticle, resulting in mechanical damage? How does worm innate immunity influence the development of P? Do interventions that affect lifespan (e.g. insulin/IGF-1 signaling and DAF-16/FoxO) act by altering mechanical stress resistance in the pharyngeal cuticle, or immune defense against bacterial invasion in the pharynx? Do other aspects of senescence lead to escape of contained infections in the pharynx in later life, and if so how? What is the cause of pharyngeal atrophy leading to p? We hope eventually to be able to provide answers to at least some of these questions.

Reviewer #3 (Remarks to the Author):

In this manuscript the authors de-convolute different causes of mortality and show that some of the more puzzling aspects of lifespan data and the resulting lifespan curves can be explained by separating the population by their mode of death. This is a very original insight about a phenomenon we all have observed but were not able to explain. The data are solid and the authors have increased the numbers provided with the last version.

In the initial manuscript I had only one major concern, which was the testing of a strain/condition with increased stress or susceptibility to infection in the pharynx. By testing ced-1 mutants, the authors have provided an excellent example.

In my view, there is additional insight that the paper provides, which is of importance but is currently hidden. By analyzing P and p deaths for animals raised in different environmental conditions (automation, NGM), showing that the frequency of P death is environment dependent, the authors also provide an explanation on why some lifespan extension mechanisms only extend lifespan under some, but not other conditions. If a lifespan mechanism prevents a form of death that does not occur in a given environment, it will not increase lifespan.

I am in full support publishing the manuscript

Authors' response: We thank the reviewer for their contribution.